# Minimum Dietary Diversity for Women of Reproductive Age (MDD-W) Data Collection: Validity of the List-Based and Open Recall Methods as Compared to Weighed Food Record

**DOI:** 10.3390/nu12072039

**Published:** 2020-07-09

**Authors:** Giles T. Hanley-Cook, Ji Yen A. Tung, Isabela F. Sattamini, Pamela A. Marinda, Kong Thong, Dilnesaw Zerfu, Patrick W. Kolsteren, Maria Antonia G. Tuazon, Carl K. Lachat

**Affiliations:** 1Department of Food Technology, Safety and Health, Faculty of Bioscience Engineering, Ghent University, 9000 Ghent, Belgium; giles.hanleycook@ugent.be (G.T.H.-C.); patrick.kolsteren@ugent.be (P.W.K.); 2Nutrition and Food Systems Division (ESN), Food and Agriculture Organization of the United Nations (FAO), 00153 Rome, Italy; jiyenalexandra.tung@fao.org (J.Y.A.T.); isabela.sattamini@fao.org (I.F.S.); mariaantonia.tuazon@fao.org (M.A.G.T.); 3Department of Food Science and Nutrition, School of Agricultural Sciences, University of Zambia, P.O. Box 32379 Lusaka, Zambia; pamela.marinda@unza.zm; 4Department of Food Science, Faculty of Agro-Industry, Royal University of Agriculture, P.O. Box 2696 Phnom Penh, Cambodia; kthong@rua.edu.kh; 5Food Science and Nutrition Research Directorate, Ethiopian Public Health Institute, P.O. Box 1242 Addis Ababa, Ethiopia; dilnesaw2012@gmail.com

**Keywords:** Cambodia, Ethiopia, list-based recall, open recall, minimum dietary diversity for women, weighed food record, Zambia

## Abstract

Minimum dietary diversity for women of reproductive age (MDD-W) was validated as a population-level proxy of micronutrient adequacy, with indicator data collection proposed as either list-based or open recall. No study has assessed the validity of these two non-quantitative proxy methods against weighed food records (WFR). We assessed the measurement agreement of list-based and open recall methods as compared to WFR (i.e., reference method of individual quantitative dietary assessment) for achieving MDD-W and an ordinal food group diversity score. Applying a non-inferiority design, data were collected from non-pregnant women of reproductive age in Cambodia (*n* = 430), Ethiopia (*n* = 431), and Zambia (*n* = 476). For the pooled sample (*n* = 1337), proportions achieving MDD-W from both proxy methods were compared to WFR proportion by McNemar’s chi-square tests, Cohen’s kappa, and receiver operating characteristic (ROC) analysis. Ordinal food group diversity (0–10) was compared by Wilcoxon matched-pairs signed-rank tests, intraclass correlation coefficients (ICC), and weighted kappa. MDD-W food groups that were most frequently misreported (i.e., type I and II errors) by the proxy methods were determined. Our findings indicate statistically significant differences in proportions achieving MDD-W, ordinal food group diversity scores, and ROC curves between both proxy methods and WFR (*p* < 0.001). List-based and open recall methods overreported women achieving MDD-W by 16 and 10 percentage points, respectively, as compared to WFR (proportion achieving MDD-W: 30%). ICC values between list-based or open recall and WFR were 0.50 and 0.55, respectively. Simple and weighted kappa values both indicated moderate agreement between list-based or open recall against WFR. Food groups most likely to be misreported using proxy methods were beans and peas, dark green leafy vegetables, vitamin A-rich fruit and vegetables, and other fruits. Our study provides statistical evidence for overreporting of both list-based and open recall methods for assessing prevalence of MDD-W or ordinal food group diversity score in women of reproductive age in low- and middle-income countries. Operationalizing MDD-W through qualitative recall methods should consider potential trade-offs between accuracy and simplicity.

## 1. Introduction

Low quality, non-diverse diets are responsible for the greatest burden of morbidity and mortality worldwide, in particular for nutritionally vulnerable women and children in low- and middle-income countries (LMICs) [1]. Monotonous diets, often dominated by starchy staples and a lack of fruits, vegetables, and animal-source foods, are the norm in resource-poor settings [2]. Nevertheless, comprehensive data on dietary patterns, diet quality, and subsequent micronutrient adequacy from nationally representative studies are scarce.

Even though numerous methods are available for assessing individual dietary intake [3], most necessitate highly proficient enumerators and exceptionally resource-intensive data collection, processing, and analysis. Many dietary assessment methodologies also require the availability of complete food composition tables, the development of which is also resource-intensive. Therefore, there is a strong and rising demand for simple and feasible, yet accurate and precise, proxy indicators to reflect micronutrient adequacy and overall diet quality [4]. The lack of indicators to allow for assessment, advocacy, and accountability has been acknowledged as a key constraint to programmatic and policy action to improve women’s diet quality [5].

In response to this demand, a dichotomous (≥5 of 10 defined food groups) minimum dietary diversity for women of reproductive age (MDD-W) indicator was developed and validated as a proxy measure for assessing micronutrient adequacy from the diet in non-pregnant women of reproductive age (WRA) at the population-level [6]. The MDD-W indicator was designed for settings where a weighed food record (WFR) or other quantitative dietary assessment methods are unfeasible. Current MDD-W guidance [7] identified two data collection approaches, namely list-based or open recall. Neither proxy method requires estimation of portion sizes (the quantity of food or drink consumed), although careful considerations are made with regard to the ≥15 g threshold for a food group to “count” in the score [7].

There are, nonetheless, remaining questions with regard to the most accurate, precise, and simplest approach for collecting MDD-W data. Both the list-based and open recall methods [7] have certain operational advantages and disadvantages for measuring dietary diversity. The list-based method requires fewer enumerator capacity requirements and shorter training time; however, it might be more likely to result in misclassification of foods and beverages into food groups or misreporting of some food items, particularly those consumed in trivial quantities [8,9]. The non-quantitative open recall method has previously been proposed, as it might produce a more accurate and complete recall of all foods and drinks consumed; however, it requires longer training time and more proficient enumerators who have a reasonable knowledge of local foods, beverages, and recipes [3,7].

However, comparability of the list-based and open recall methods against WFRs (i.e., reference method of individual quantitative dietary assessment) has not been investigated in the context of evaluating the performance of MDD-W data collection. To our knowledge, one study has examined the relative validity of the list-based and open recall methods against each other, but not compared to WFR, to assess MDD-W in pregnant women in Bangladesh and India [8]. Furthermore, Martin–Prével et al. (2010) compared simple food group diversity indicators derived from list-based recall to the same indicators derived from quantitative 24-h recalls [9].

In this study, we address the research gap by using comparable dietary intake data from non-pregnant WRA in Cambodia, Ethiopia, and Zambia to examine the measurement agreement of list-based and open recall methods as compared to WFR for predicting MDD-W, ordinal food group diversity scores, and individual food group consumption.

## 2. Materials and Methods

Our research was reported using the STROBE-nut checklist [10].

### 2.1. Sample Size

The key aim of this multi-country study was to determine whether proxy methods are comparable to WFR at capturing the proportion of WRA achieving MDD-W. Non-inferiority tests of correlated proportions were designed for this purpose. We used PASS 16 Power Analysis and Sample Size Software (2018) to determine the sample size, using the following assumptions [11]: 90% power (1-*β*), 5% significance level (*α*), 5% maximum allowable difference between proxy methods and WFR, 0% actual difference, 56% (Cambodia); 13% (Ethiopia); 57% (Zambia) standard MDD-W proportion, and 10% nuisance parameter. Our initial calculations resulted in a sample size of 374 WRA. However, accounting for a 15% dropout rate, the final sample size was 430 WRA per dietary assessment method in each country. This study pooled data from Cambodia, Ethiopia, and Zambia (*n* = 1337) as <15% of the associations between MDD-W recalled from proxy methods and WFR were explained by country-level clustering when fitting mixed effects logistic regression models (random intercept: Country; Appendix A).

### 2.2. Data Sources and Study Population

The study countries were selected based on their diverse geographic locations, variation in dietary patterns, and the consideration that they are part of the Deutsche Gesellschaft für Internationale Zusammenarbeit’s (GIZ) special initiative One World—No Hunger program, which collected MDD-W survey data among WRA in 12 countries [12]. Thus, our multi-country study was able to reference GIZ’s country-specific experience and survey tools during the preparatory stage.

In Cambodia, data were collected in the rainy season between August and September 2019 in Kampong Thom Province in 34 villages (*phum*) in 5 rural communes (*khum*) from 2 districts (*srok*). In Ethiopia, data were collected in the rainy season between June and August 2019 in the Amhara Region in 24 villages (*gotes*) in 13 sub-districts (*kebeles*) from 3 districts (*woredas*), characterized by rural, peri-urban, and urban residencies. In Zambia, data were collected in the hot and dry/wet seasons between October and December 2019 in the Chongwe District in Lusaka Province in 6 villages from 3 peri-urban sub-districts (*wards*). Convenience samples of non-pregnant WRA (15–49 years) were selected from each study site in Cambodia (*n* = 430) and Zambia (*n* = 476), whereas representative samples (i.e., probability proportional to size) were selected in Ethiopia (*n* = 431).

### 2.3. Ethical Standards Disclosure

The studies received ethical approvals from the National Ethics Committee for Health Research (RUA/NECHR/053/2019) in Cambodia, Ethiopian Public Health Institute Scientific and Ethical Review office (EPHI/IRB/156/2018) in Ethiopia, and the University of Zambia Biomedical Research Ethics Committee (UNZA/IRB/304/2019) in Zambia. Written informed consent was obtained from all study participants.

### 2.4. Preparatory Phase

Prior to data collection in Cambodia, Ethiopia, and Zambia, surveys and, in particular, food lists were adapted to the local context (i.e., focus group discussions with nutritionists and health workers), based on existing country-specific MDD-W data collection tools [12]. The adapted surveys were programmed into tablets, using the open data kit (ODK) platform (https://opendatakit.org), in both English and the local language.

One-week capacity development workshops were held for MDD-W data collection teams, including classroom training and a field practice (i.e., pretest) of all three dietary assessment methods, using the tablet-based surveys. Furthermore, a pilot study was conducted on at least 50 WRA in each study country, to further improve the data collection tools, data quality checks, data management, statistical analysis procedures, and enumerator confidence and capacity (between May and August 2019).

### 2.5. Data Collection

We used quantitative WFR and qualitative list-based and open recall, as described in FAO and FHI 360 (2016), to collect individual data on the intake of distinct food groups over a 24-h period [7]. The proxy methods were both enumerated on the day after the WFR, but by different interviewers and at different times (both at random, using the MS Excel RAND function in Ethiopia and Zambia and the random generator application in Cambodia). Across all three study countries, no attrition was observed and no data were excluded (Figure 1).

To conduct the WFR, each interviewer trained in the WFR method spent the entire day (i.e., from the time the first food or beverage was consumed in the morning (±6 am) to when the last one was consumed at night (±9 pm)) in a chosen household weighing and recording all foods and beverages consumed by the participant using digital scales accurate to ± 1 g and calibrated daily. Participants were instructed not to change their normal dietary pattern and to serve their own portions of food on separate utensils from other household members on the WFR day. On the occasions when subjects attended events outside the home, the interviewer accompanied them to weigh and record any food or beverages consumed [13,14].

Detailed weighed recipe data were also collected for all the composite dishes consumed during the WFR. These data were then used to calculate the weight (g) of each ingredient consumed by respondents within mixed dishes, after adjustments, where necessary, for preparation yield using US Department of Agriculture (USDA) factors [15], because there are no Cambodian, Ethiopian, or Zambian data.

During the list-based recall, the trained enumerator read a list of foods and drinks from each food group, displayed on a tablet with the ODK platform, to the respondent. A priori, local nutritionists reached a consensus on foods that would be consumed in daily quantities ≥15 g threshold. The non-quantitative questionnaire was based on a list of 18 food groups in Ethiopia, and 19 food groups in Cambodia and Zambia (Appendix A) that reflected the unique characteristics of food consumption in each country (e.g., including the optional “insects and other small protein foods” category for the latter). Various examples of local foods or composite dishes made from these local foods were provided for each group.

The open recall method evaluated food and drink consumption using an ODK tablet-based multiple pass qualitative 24-h recall. WRAs were requested to describe all the foods and drinks they consumed during the preceding day and night, as well as the meal time. Recipes of mixed dishes were enumerated by asking the women who had prepared and consumed them to recall all the ingredients they added. Foods and ingredients usually consumed daily in less than a tablespoon quantity (i.e., <15 g) were defined during country-specific survey adaption and enumerators were trained on the principle of not counting them towards the MDD-W indicator before data collection, as described in the MDD-W measurement guide [7].

### 2.6. Constructing MDD-W

For the WFR, the food and drink items were recategorized into the 10 food groups used in the MDD-W measurement guide [7]: (1) Starchy staple foods; (2) beans and peas; (3) nuts and seeds; (4) dairy products (milk, yoghurt, and cheese); (5) flesh foods (meat, fish, poultry, and liver/organ meats); (6) eggs; (7) dark green leafy vegetables; (8) vitamin A-rich fruits and vegetables; (9) other vegetables; and (10) other fruits. A WFR food group diversity indicator (0–10) was obtained by summing the number of food groups consumed in a daily quantity of at least 15 g by each WRA.

For the list-based recall, the food groups were aggregated into the 10 defined MDD-W food groups. In the qualitative open recall, all food items were assembled into the identical 10 food groups. These food group variables were used for descriptive analyses of misreporting in recalled consumption of the food group as compared to WFR. For the proxy methods, food group diversity scores (0–10) were constructed by summing the number of food groups consumed by each WRA.

### 2.7. Statistical Analysis

First, we tested the effect of allocation of proxy methods (day 2) on MDD-W (i.e., potential “carry over” effect [16,17]), using a two-sample test of proportions for the dichotomous variable and Wilcoxon rank-sum test of the difference in food group diversity scores. We used descriptive analyses to report the proportion of WRA reaching MDD-W, median food group diversity score, and individual food group consumption.

For our main outcome of interest, we performed McNemar’s chi-square tests for paired proportions to assess how well both proxy methods estimated MDD-W and individual food group consumption compared to WFR. To measure agreement amongst WRA achieving MDD-W, we used simple Cohen’s kappa statistics. Kappa scores of 0.21–0.40 indicate fair agreement; 0.41–0.60, moderate agreement; 0.61–0.80, substantial agreement; 0.81–1.00, almost perfect agreement [18]. We also performed a “gold standard” receiver operator characteristic (ROC) analysis to compare how well the list-based and open recall methods predicted MDD-W as compared to WFR. The area under the curve (AUC) summarizes the predictive power of MDD-W for the proxy methods. An AUC significantly different from 0.5 and ≥0.70 was deemed satisfactory to indicate accuracy [8].

Furthermore, as the 10-point ordinal score might be preferred in research and certain programmatic contexts [6], we also performed Wilcoxon matched-pairs signed-rank test, intraclass correlation coefficients (ICC), and weighted Cohen’s kappa to assess how well both proxy methods estimated food group diversity score compared to WFR. The ICC was used to estimate the consistency between dietary assessment methods with a higher ICC representing a higher degree of consistency [19,20]. We calculated the correlation of proxy method against WFR using Spearman’s rank correlation coefficient, which is fitting, since we considered our food group diversity variables to be ordinal.

We quantified the frequency of misreporting (i.e., type I and II errors) for MDD-W and each individual food group using confusion matrices and identified the food groups that were most often misreported by the proxy methods.

Data management and statistical analysis were conducted in Stata version 15.1 [21]. A two-sided significance level of *p* < 0.05 was applied for all analyses.

## 3. Results

The allocation sequence of proxy methods was non-significant for the dichotomous MDD-W indicator (*p* = 0.64) and ordinal food group diversity score (*p* = 0.49). Therefore, our subsequent analyses included all study data from list-based and open recall methods (both *n* = 1337).

### 3.1. Sample Characteristics

Most non-pregnant WRA (>60%) were married with a mean ± SD age ranging between 36 ± 8 years in Cambodia (*n* = 430) and 29 ± 9 years in Zambia (*n* = 476). In Cambodia almost all WRA were Buddhist (>90%), whereas in Ethiopia and Zambia large proportions of WRA were Christians (>60%). In all three study counties >50% of WRA were the wives of a male household head, >40% were homemakers by occupation, and only 3% were formally employed. Our WFR findings showed that in Cambodia, 44% of WRA achieved MDD-W, 8% in Ethiopia (*n* = 431), and 38% in Zambia. The median (interquartile range (IQR)) food group diversity score in Ethiopia (3 (3,3)) was 1 food group lower than in Cambodia (4 (4,5)) and Zambia (4 (4,5)), based on WFR.

### 3.2. Comparison of the Performance of Proxy Methods against WFR to Predict Dichotomous MDD-W

The proportion (95% confidence interval (CI)) of WRA achieving MDD-W (consumed ≥5 food groups; *n* = 1337) was 30% (95% CI: 28–33) as stated by the WFR, whereas according to the list-based recall it was 46% (95% CI: 43–48), and 40% (95% CI: 37–42) using the open recall method (Table 1; both *p* < 0.001). The list-based and open recall methods correctly classified (i.e., true positive and negative values) 77% and 80% of WRA as (not) achieving MDD-W, respectively (Table 2). Simple kappa values for the dichotomous indicator indicated moderate agreement for both proxy methods against WFR (k = 0.51–0.57). Furthermore, list-based recall had a sensitivity (i.e., truly achieving MDD-W) of 87% and a specificity (i.e., truly not achieving MDD-W) of 72%, whereas open recall had a sensitivity and specificity of 83% and 79%, respectively (Figure 2). The pooled performance of proxy methods to predict dichotomous MDD-W, compared to WFR (AUC: 1), was comparable for the list-based and open recall methods (AUCs ranged from 0.68 to 0.70 in Cambodia, 0.89 to 0.91 in Ethiopia, and 0.76 to 0.80 in Zambia). Nevertheless, the pooled AUCs for the list-based and open recall methods were significantly different from those of WFR (both *p* < 0.001).

### 3.3. Measurement Agreement between List-Based or Open Recall and WFR for Food Group Diversity Score

The distributions of food group diversity scores, by dietary assessment method, are presented in Appendix A. The pooled median (IQR) number of WFR food groups was 4 (3,5). Ordinal food group diversity scores for the list-based and open recall were statistically higher than those of WFR (both *p* < 0.001; Table 3). The Spearman correlations (ρ) between WFR and proxy methods were 0.67 and 0.70 for the list-based and open recall, respectively (both *p* < 0.001).

ICC values ranged from 0.27 to 0.28 in Cambodia, 0.55 to 0.60 in Ethiopia, and 0.43 to 0.52 in Zambia for list-based and open recall, respectively. The agreement of food group diversity score was 47% for list-based and 52% for open recall. Weighted kappa values for the ordinal food group diversity score (k = 0.47–0.52) were very similar to the simple kappa for the dichotomous MDD-W in all three countries.

### 3.4. Misreporting of Food Groups by List-Based and Open Recall as Compared to the Reference Method

For the list-based recall, 83% of the measurement errors were type I (false positive values); as compared to 74% for open recall (Table 2). Our individual MDD-W food group confusion matrices indicated that for the list-based recall, 69% of type I errors arose from overreporting of beans and peas (11%), dark green leafy vegetables (20%), vitamin A-rich fruits and vegetables (12%), and other fruit (25%) food groups, whereas 31% of type II errors (false negative values) were from underreporting vitamin A-rich fruits and vegetables (Table 4). In parallel for open recall, 68% of type I errors came from overreporting of beans and peas (14%), dark green leafy vegetables (25%), vitamin A-rich fruits and vegetables (11%), and other fruit (19%) food groups, whereas 32% of type II errors were from underreporting vitamin A-rich fruits and vegetables.

Several food groups that made up the dichotomous MDD-W indicator and food group diversity score varied significantly (*p* < 0.05) when enumerated by the list-based and open recall methods (Table 1). In Cambodia, comparing proxy methods with WFR, our findings indicate overreporting of beans and peas (∼6 percentage points (pp)); nuts and seeds (5–9 pp); dark green leafy vegetables (∼27 pp); and other fruits (23 pp for list-based recall). In Ethiopia, similarly, respondents overreported beans and peas (by∼10 pp); dark green leafy vegetables (2–5 pp); and other fruits (2–7 pp) using the list-based and open recall. In Zambia, the picture was slightly different: When comparing proxy methods to WFR, our findings indicate overreporting of flesh foods (8–13 pp); dark green leafy vegetables (9 pp); and other fruits (17–21 pp; Appendix A). Overall, the list-based method reported higher proportions of MDD-W food group consumption than the open recall.

## 4. Discussion

Our findings indicate poor dietary diversity in surveyed non-pregnant WRA from Cambodia, Ethiopia, and Zambia with only 44% (95% CI: 39–49), 8% (95% CI: 5–11), and 38% (95% CI: 33–42) achieving MDD-W, respectively (based on WFR). Both the list-based and open recall methods did not meet our non-inferiority criterion in predicting dichotomous MDD-W (i.e., our pooled results indicate >5% difference as compared to WFR). Nevertheless, AUCs were 0.79 for list-based and 0.81 for open recall, which were both considered satisfactory for predictive performance (although statistically different from WFR).

Food group diversity scores measured by the proxy methods had moderate agreement, but were also statistically different when compared to WFR (although median (IQR) were identical). In all countries, both the list-based, in particular, and open recall were inclined to misreport consumption of specific MDD-W food groups. To illustrate, overreporting for both proxy methods as compared to WFR exceeded 10% for beans and peas, nuts and seeds, dairy, dark green leafy vegetables, and other fruits.

We attempted to identify the specific foods and beverages (and/or potential recall biases) that were accountable for the misreporting of food group consumption by looking in more detail at the list-based and open recall, as compared to the WFR data. The overreporting of food groups with list-based and open recall are, in part, due to the reporting of food items not consumed in sufficient quantities (≥15 g; e.g., nuts and seeds often used as “condiments and seasonings”). This is consistent with findings from Bangladesh and India [8] and Burkina Faso [9], where misreporting was common for ingredients used in composite dishes or in small quantities for sauces. The overreporting of certain food groups not consumed according to the reference method could be a result of memory lapses, social desirability, and/or social approval bias e.g., flesh foods, dairy, eggs, certain sorts of vegetables [3]. Our country-level findings will feed back into future decisions made during preparatory phases to improve qualitative MDD-W data collection tools.

The poorer agreement and lower correlation among proxy methods and WFR in Cambodia as compared to Ethiopia and Zambia need further examination. In all three countries, we conducted theoretical and practical enumerator capacity development workshops on dietary assessment methodologies, with a standardized adaption of questionnaires. The nature of the diet in Cambodia is the most complex, with numerous mixed dishes containing a large number of ingredients. Nevertheless, special focus was placed during the enumerator training to make sure that composite dishes and foods items with multiple ingredients were allocated in the correct food groups. As previously described, ingredients with trivial quantities mainly used to add flavor were to be classified in the “condiments and seasonings” food group.

The *Minimum Dietary Diversity for Women: A Guide to Measurement* describes the use of two proxy qualitative methods (i.e., list-based or open recall) to measure food group diversity [7], however no evidence existed on the validity of these two non-quantitative methods at the time of publication. Our multi-country study used WFR, which is accepted as the reference method for dietary assessment [3]. At present, however, there is no true gold standard for dietary intake, since all methodologies entail error. Therefore, our comparisons of the dichotomous MDD-W indicator and ordinal food group diversity score calculated from list-based and open recall against WFR might still be considered as a type of relative validity. Both the list-based and open recall methods in our study covered the same period of recall (i.e., the day and night prior to the survey, on which WFR was conducted), thereby preventing differences related to varying time frames.

To the authors’ knowledge, this is the first multi-country study that uses dietary intake data from countries in Sub-Saharan Africa and Southeast Asia, with a wide range of food cultures, to assess the validity of two proxy methods to predict MDD-W as compared to WFR for non-pregnant WRA. Strengths of the study include that we compared the two qualitative methodologies described for MDD-W against WFR (i.e., the reference method of individual quantitative dietary assessment). Furthermore, our study collected data, with high-quality capacity development and methods, resulting in high-quality dietary intake data, with sufficiently large sample sizes per country. The present findings are timely, given that there is a widespread uptake of MDD-W (e.g., Demographic and Health Surveys-8 [22]) and strong support to scale-up the indicator at country-level from the European Union, United Nations agencies, and international NGOs (e.g., national information systems and nutrition-sensitive agricultural programs).

Qualitative appraisal of the simple Cohen’s kappa and AUC results showed similar reliability and accuracy between the list-based and open recall, demonstrating that list-based data collection performed almost as well as open recall in estimating proportions of WRA achieving MDD-W. These findings have important implications, because the list-based recall has several operational advantages, such as lower enumerator capacity requirements and shorter training time, which in turn might reduce the cost of MDD-W data collection. Nevertheless, our findings confirm statistical inferiority (i.e., overreporting) of both list-based and open recall, as simple tools to measure dietary diversity in WRA. Furthermore, for assessing the prevalence specific MDD-W food group consumption, estimates from list-based and open recall differed quite substantially for distinct food groups in each country, so additional methodological work from a wider range of (urban) contexts and seasons is required to understand the potential of using simple proxy methodologies to assess the consumption of individual food groups.

This research was part of a Food and Agriculture Organization of the United Nations-led project that will result in updated operational guidance for measuring MDD-W. The guidance will also be informed by operational data, which will factor into a recommendation on the choice of MDD-W data collection method.

## Figures and Tables

**Figure 1 nutrients-12-02039-f001:**
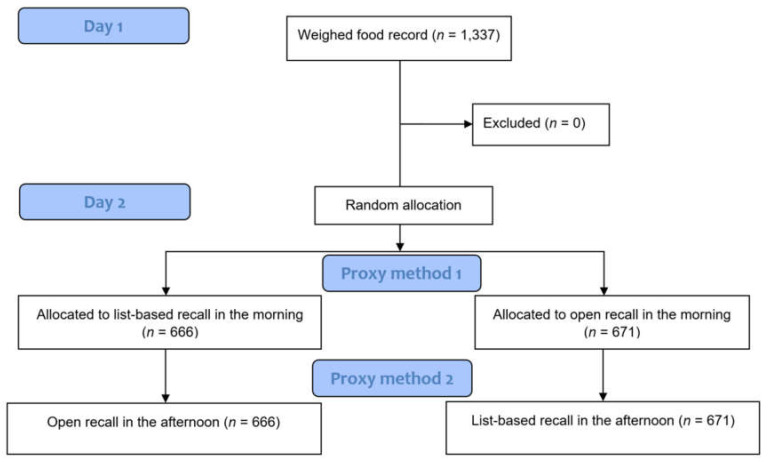
Study design diagram.

**Figure 2 nutrients-12-02039-f002:**
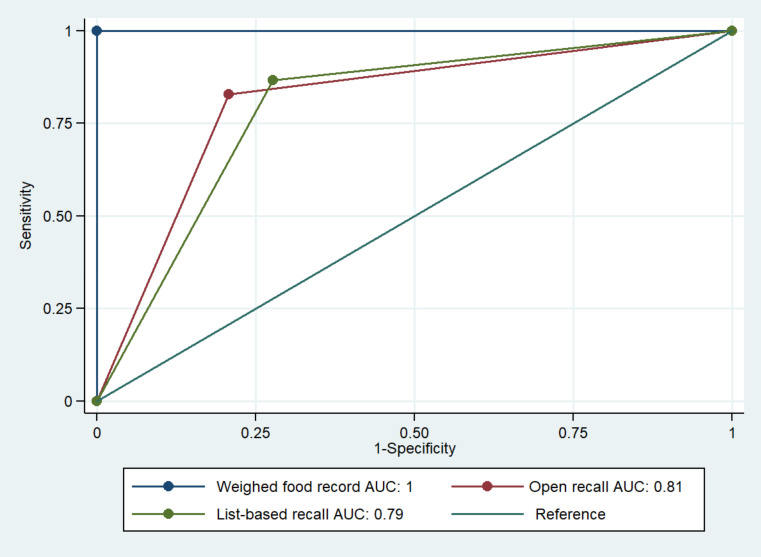
Receiver operating characteristic analysis for dichotomous minimum dietary diversity for women of reproductive age measured by list-based and open recall methods as compared to weighed food record. AUC, area under the curve.

**Table 1 nutrients-12-02039-t001:** Pooled proportions of non-pregnant women (15–49 years) having consumed food groups and achieved minimum dietary diversity for women of reproductive age, based on weighed food record, list-based, and open recall methods.

	Weighed Food Record (*n* = 1337)	List-Based Recall (*n* = 1337)	Open Recall (*n* = 1337)
All starchy staple foods	100	99.6	100
Beans and peas	41.6	46.8 ***	46.8 ***
Nuts and seeds	4.2	8.8 ***	7.3 ***
Dairy	2.9	6.6 ***	6.1 ***
Flesh foods	48.5	53.9 ***	52.3 ***
Egg	22.1	23.4 *	21.5
Dark green leafy vegetables	40.4	53.7 ***	53 ***
Vitamin A-rich fruits and vegetables	26.4	28.3	23.6 **
Other vegetables	92.1	92.2	92.6
Other fruits	14.6	31.3 ***	22.1 ***
MDD-W	30.1	45.5 ***	39.5 ***

Statistically different proportions between weighed food record and proxy methods are indicated by * *p* < 0.05, ** *p* < 0.01, and *** *p* < 0.001. MDD-W, minimum dietary diversity for women of reproductive age.

**Table 2 nutrients-12-02039-t002:** Agreement between dichotomous minimum dietary diversity for women of reproductive age measured by list-based or open recall as compared to weighed food record.

	Weighed Food Record		Agreement Statistics	
	<5 Food Groups *n* (%)	≥5 Food Groups *n* (%)	% Agreement	Cohen’s Kappa
**List-Based Recall**				
<5 food groups	675 (50.5)	54 (4) ^‡^	76.6	0.51 ***
≥5 food groups	259 (19.4) ^†^	349 (26.1)		
**Open Recall**				
<5 food groups	740 (55.3)	69 (5.2) ^‡^	80.3	0.57 ***
≥5 food groups	194 (14.5) ^†^	334 (25)		

^†^ False positive finding (type I error). ^‡^ False negative finding (type II error). *** *p* < 0.001.

**Table 3 nutrients-12-02039-t003:** Agreement between ordinal food group diversity score measured by list-based or open recall as compared to weighed food record.

	Median (IQR Range) ^†^	ICC (95% CI)	% Agreement	Weighted Kappa
**Weighed food record (reference)**	4 (3, 5)	-	-	-
**List-based recall**	4 *** (3, 5)	0.50 (0.15, 0.85)	47.3	0.47 ***
**Open recall**	4 *** (3, 5)	0.55 (0.18, 0.92)	52.2	0.52 ***

^†^ Wilcoxon matched-pairs signed-rank test. *** *p* < 0.001. CI, confidence interval; ICC, intraclass coefficients; IQR, interquartile range.

**Table 4 nutrients-12-02039-t004:** Agreement between 10 minimum dietary diversity for women of reproductive age food groups measured by list-based or open recall as compared to weighed food record.

	Weighed Food Record		Agreement Statistics	
	<15 g *n* (%)	≥15 g *n* (%)	% Agreement	Cohen’s Kappa
	**All Starchy Staple Foods**			
**List-Based Recall**				
<15 g	0 (0)	5 (0.4) ^‡^	99.6	1
≥15 g	0 (0) ^†^	1332 (99.6)		
**Open Recall**				
<15 g	0 (0)	0 (0) ^‡^	100	-
≥15 g	0 (0) ^†^	1337 (100)		
	**Beans and Peas**			
**List-Based Recall**				
<15 g	666 (49.8)	46 (3.4) ^‡^	88	0.76 ***
≥15 g	115 (8.6) ^†^	510 (38.1)		
**Open Recall**				
<15 g	668 (50)	43 (3.2) ^‡^	88.3	0.76 ***
≥15 g	113 (8.5) ^†^	513 (38.4)		
	**Nuts and Seeds**			
**List-Based Recall**				
<15 g	1199 (89.7)	20 (1.5) ^‡^	92.4	0.38 ***
≥15 g	82 (6.1) ^†^	36 (2.7)		
**Open Recall**				
<15 g	1222 (91.4)	17 (1.3) ^‡^	94.3	0.48 ***
≥15 g	59 (4.4) ^†^	39 (2.9)		
	**Dairy**			
**List-Based Recall**				
<15 g	1247 (93.3)	2 (0.1) ^‡^	96	0.57 ***
≥15 g	51 (3.8) ^†^	37 (2.8)		
**Open Recall**				
<15 g	1252 (93.6)	4 (0.3) ^‡^	96.3	0.57 ***
≥15 g	46 (3.4) ^†^	35 (2.6)		
	**Flesh Foods**			
**List-Based Recall**				
<15 g	610 (45.6)	7 (0.5) ^‡^	93.6	0.87 ***
≥15 g	78 (5.8) ^†^	642 (48)		
**Open Recall**				
<15 g	624 (46.7)	14 (10.5) ^‡^	94.2	0.88 ***
≥15 g	64 (4.8) ^†^	635 (47.5)		
	**Egg**			
**List-Based Recall**				
<15 g	993 (74.3)	31 (2.3) ^‡^	94	0.83 ***
≥15 g	49 (3.7) ^†^	264 (19.7)		
**Open Recall**				
<15 g	1007 (75.3)	43 (3.2) ^‡^	94.2	0.83 ***
≥15 g	35 (2.6) ^†^	252 (18.8)		
	**Dark Green Leafy Vegetables**			
**List-Based Recall**				
<15 g	587 (43.9)	32 (2.4) ^‡^	81.9	0.64 ***
≥15 g	210 (15.7) ^†^	508 (38)		
**Open Recall**				
<15 g	593 (44.4)	36 (2.7) ^‡^	81.7	0.65 ***
≥15 g	204 (15.3) ^†^	504 (37.7)		
	**Vitamin-A Rich Fruits and Vegetables**			
**List-Based Recall**				
<15 g	857 (64.1)	102 (7.6) ^‡^	82.9	0.57 ***
≥15 g	127 (9.4) ^†^	251 (18.8)		
**Open Recall**				
<15 g	894 (66.9)	128 (9.6) ^‡^	83.7	0.57 ***
≥15 g	90 (6.7) ^†^	225 (16.8)		
	**Other Vegetables**			
**List-Based Recall**				
<15 g	47 (3.5)	57 (4.3) ^‡^	91.3	0.40 ***
≥15 g	59 (4.4) ^†^	1174 (87.8)		
**Open Recall**				
<15 g	42 (3.1)	57 (4.3) ^‡^	90.9	0.36 ***
≥15 g	64 (4.8) ^†^	1174 (87.8)		
	**Other Fruits**			
**List-Based Recall**				
<15 g	888 (66.4)	31 (2.3) ^‡^	78.7	0.42 ***
≥15 g	254 (19) ^†^	164 (12.3)		
**Open Recall**				
<15 g	985 (73.7)	57 (4.3) ^‡^	84	0.47 ***
≥15 g	157 (11.7) ^†^	138 (10.3)		

^†^ False positive finding (type I error). ^‡^ False negative finding (type II error). *** *p* < 0.001.

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
