# Peer review of "Minimum Dietary Diversity for Women of Reproductive Age (MDD-W) Data Collection: Validity of the List-Based and Open Recall Methods as Compared to Weighed Food Record"

_nutrients, 2020, doi:10.3390/nu12072039_

Round 1

Reviewer 1 Report

 Minimum Dietary Diversity for Women of Reproductive Age (MDD-W) Data Collection: Validity of the List-Based and Open Recall Methods as Compared to Weighed Food Record

This paper describes the performances of an indicator (MDD-W) calculated from two data collection methods (list-based or qualitative open recalls) to predict the same indicator calculated from weighed food records (WFR). The study is important, as evidence is missing on how to best collect dietary data to compile MDD-W; while MDD-W is widely used by a wide range of actors to monitor micronutrient adequacy of women’s diets in low-income countries. Strengths of the study design include the multi-country design and the randomization of the ordering of recalls. Statistical methods are appropriate, and the manuscript is clear and well written. However, I have three concerns about this paper which, if addressed, would greatly help in convincing the reader that the conclusions are sufficiently supported by results. These concerns are related to 1) the strong focus made on WFR as the gold standard for dietary data collection, with insufficient description of WFR methods ; 2) the absence of statistical comparison between the two proxy indicators tested ; and 3) the absence of analysis of  the performance of the tested indicators to predict micronutrient adequacy, while it would seem possible to calculate micronutrient adequacy from WFR, and MDD-W is a proxy for micronutrient adequacy.

My first major comment relates to the presentation of WFR as the gold standard in terms on dietary data collection. This should be nuanced, which would lead to nuancing the conclusion as well. My point is that it is debatable that WFR is/should be the gold standard for dietary assessment. In addition, MDD-W validation studies used 24HR data, meaning that WFR is not either a gold standard to construct MDD-W indicators.

I provide below an explanation on why it seems to me that it would it be possible that WFR underreported food intakes. The authors should either be much more nuanced when interpreting the discrepancy between WFR and qualitative data collection methods ; or provide many more details in the methods section on the way WFR were handled, to justify with little possible doubt that this is WFR which captured the best true food consumption.

Specifically, on line 332, you stated: “Our study used WFR, which is accepted as the reference method for dietary assessment (3).” I carefully read again the reference you provided to justify this statement (Gibson et al, 2017), and I would not conclude that Gibson et al present WFR as the reference method for dietary assessment. You recognized right after that each methodology entail error, but you did not discuss this further. It would be useful to list what kind of error might be expected from WFR, as well as its advantages. Advantages include the fact that the food groups consumed were observed (during the time the enumerator shadowed the respondent), and that the 15g cut-off (for a food group to count) was precisely measured, and not estimated. One obvious limitation is that several consumptions might be missed by enumerators when they cannot spend a full 24hr in the household. Actually, on lines 336-337, you stated that the recall period for both recalls was “the day and night prior to the survey, on which WFR was conducted”. However, on line 151, it is stated that “To conduct the WFR, each interviewer trained in the WFR method spent the entire day in a chosen household”. Therefore, it does seem that the recall periods were, in fact, different, and that the WFR did not observe overnight and, possibly, late evening food consumptions. If the night included in the recall period for the qualitative questionnaires was the night just before the recall (which it usually is when performing 24hr recalls), then any food consumed late in the evening or during the night after the WFR enumerator left the household could not possibly be captured by the WFR enumerator. On the other hand, these late-evening and overnight consumptions would be captured with both qualitative recalls… In this case, the WFR would underestimate true consumption, and would underestimate whether or not the respondent achieved MDD-W.

Related to this, the section on sample characteristics describes that a pretty large proportion of women were NOT homemaker: It would be useful to get more insight on these women’s occupation (with exact proportions, possibly by country, at least in a supplemental table) and explain on line 151 how you handled WFR in working women (who may spend the day out for work, not just for “attending events”).Also, on line 320: If the enumerators for WFR did not stay at home for a full 24 hours or were not always able to shadow women everywhere they went (eg at work), the apparent overreporting with recalls could be due to underreporting of true intake with WFR (see comment above). This could be discussed here or in the following paragraph.

My second major comment relates to the direct comparison which is made toward the end of the discussion between the 2 tested recall methods. On line 349, you stated there were “similar predictive values between the list-based and open recall”.  Unless I mis-read the results, I didn’t see any direct statistical measure of agreement between the list based and the open recall indicators, to confirm whether or not they are equivalent. What I saw, however, was that the open recall did systematically perform slightly better than the list-based questionnaire at predicting MDD-W calculated from WFR.  What made you conclude that an overestimation of 16 pp of MDD-W is similar to an overestimation of 10pp? In other words, when would we conclude that a prevalence of 46% is similar to a prevalence of 40%, when we estimate the “true” prevalence to be 30% (regardless of my comment above about whether WFR would give the true prevalence). In the absence of statistical testing to confirm whether the differences between the 2 recall methods were significant, the systematic direction of the difference in performance between the 2 indicators should at least be better acknowledged. In the meantime, would it be possible to statistically compare both proxy methods? They were obtained on the same reference period, so it should be possible.

Finally, you stated on lines 340 that the aim of the study was to compare the validity of two proxy methods as compared to WFR. This is not exactly what you did. You assessed the validity of a proxy indicator obtained from two data collection methods as compared to te same proxy indicator obtained from WFR. You should not short-cut the precision that this were the data collection methods which were compared. If you compared “proxy methods to WFR”, one would expect you to use quantitative dietary data to calculate probabilities of micronutrient adequacy, and identify the best data collection method to compile MDD-W so that it is an accurate proxy indicator of acceptable micronutrient adequacy (what MDD-W was designed to be). I am wondering what prevented you to perform this analysis (ie study performance against micronutrient adequacy)?

Minor comments

Intro line 66: “validated as a proxy measure for assessing micronutrient adequacy from the diet in non-pregnant  and non-lactating women of reproductive age (WRA) at the population-level”. To my understanding, it has also been validated for use in lactating women, in that same study.

Section sample characteristics, line 222: not sure why religion is useful here.

Table 2: please use a consistent number of significant digits or consistent number of decimals (whatever is preferred by the journal) for percentages.

Line 305: “> 5 % difference”: this is confusing. did you mean “<”, as you refer to the crierion? Or shall you word it differently such as “i.e. our results showed >5% difference…”?

Discussion section, Line 312: It would be useful to see these results, at least as supplementary material.

Discussion section, Line 316: if some condiments and seasoning were reported in qualitative recall, could it be because the preparatory work mis-identified those ingredients which would be used as condiments and seasoning? Should you comment on whether and how your results could feed-back into decisions made during preparatory work to improve the qualitative data collection tools?

Line 346: there seems to be a typo in how reference 8 is listed

Line 159: I was initially confused by this sentence. for clarity, I suggest to replace “These data were then used to calculate the weight (g) of each ingredient, after adjustments,” with something like “These data were then used to calculate the weight (g) of each ingredient eventually consumed by respondents within mixed dishes, after adjustments for cooking yield”

Line 174-175: “consumed in less than a tablespoon quantity (i.e. < 15g)”. I assume it was daily consumption? please specify. Same question for line 184 “consumed in a quantity of at least 15g by each WRA”.

Line 231: the title  “Comparison of the performance of proxy methods and WFR to predict dichotomous MDD-W” is confusing. Your standard MDD-W is the WFR one. Did you mean “Comparison of the performance of MDD-W calculated from proxy methods to predict MDD-W calculated from WFR”?

Reviewer 2 Report

I have attached my brief comments. Overall, it is an interesting publication.  

Reviewer 3 Report

The study of Hanley-Cook et al. evaluated statistical evidence for overreporting of both list-based and open recall methods for assessing prevalence of MDD-W or ordinal food group diversity score in women of reproductive age in low- and middle-income countries. The study is comprehensive and generally supports the authors' conclusions. These findings may benefit from some additional clarification, as detailed below.

Comments

  • The impact of the study should be better clarified and detailed.
  • The manuscript should be edited to correct contextual and layout errors.

Round 2

Reviewer 1 Report

The authors responded adequately to my questions and concerns. Thanks for the clarity of the answers.  I don't have further comments